# Dispositional optimism weakly predicts upward, rather than downward, counterfactual thinking: A prospective correlational study using episodic recall

Jessica Gamlin[1]*, Rachel Smallman[2], Kai Epstude[3], Neal J. Roese[4]

1 Department of Marketing, Lundquist College of Business, University of Oregon, Eugene, Oregon, United States of America, 2 Department of Psychology, Texas A&M University, College Station, Texas, United States of America, 3 Department of Psychology, University of Groningen, Groningen, Netherlands, 4 Department of Marketing, Kellogg School of Management, Northwestern University, Evanston, Illinois, United States of America

* jgamlin@uoregon.edu

**Data Availability Statement:** All deidentified data files are available at OSF at https://osf.io/wudjs/.

**Funding:** The authors received no specific funding for this work.

## Abstract

Counterfactual thoughts center on how the past could have been different. Such thoughts may be differentiated in terms of direction of comparison, such that upward counterfactuals focus on how the past could have been better, whereas downward counterfactuals focus on how the past could have been worse. A key question is how such past-oriented thoughts connect to future-oriented individual differences such as optimism. Ambiguities surround a series of past studies in which optimism predicted relatively greater downward counterfactual thinking. Our main study ($N = 1150$) and six supplementary studies ($N = 1901$) re-examined this link to reveal a different result, a weak relation between optimism and upward (rather than downward) counterfactual thinking. These results offer an important correction to the counterfactual literature and are informative for theory on individual differences in optimism.

## Introduction

### Goal of the present research

Looking back on one's past to compare what actually transpired to what might have been, (i.e., counterfactual thinking) is a common feature of mental experience [1–3]. Further, counterfactual thoughts may be differentiated in terms of their direction of comparison, where upward counterfactuals center on how an outcome could have been better than actuality and downward counterfactuals center on how an outcome could have been worse than actuality. Direction of comparison has been widely used to parse the content of counterfactual thinking. Such counterfactual thoughts about past outcomes may also connect to future-oriented individual differences such as dispositional optimism, which is defined as domain-general beliefs that future outcomes will be positive [4, 5].

**Competing interests:** The authors have declared that no competing interests exist.

A subset of the counterfactual literature (comprising nine studies in six papers [6–11]; see S1 Appendix) has examined the relationship between dispositional optimism and counterfactual direction of comparison, finding that optimism predicts downward (vs. upward) counterfactual thinking. That is, people who tend to hold positive expectancies about future outcomes tend to think about how things in the past could have been worse (rather than better). Although past studies indicate that optimism predicts downward counterfactual thinking [6–11], we identify theoretical and methodological reasons to question those prior results.

Recent trends in scholarly research, particularly within the field of psychology, illuminate the need to consider existing findings in light of new research practices and/or revised theorizing [12–15]. In particular, bias toward the publication of only "clean," significant findings has introduced a gap between reproducible effects and the refinement of existing (i.e., published) knowledge. Proposed solutions to bridge this gap include (a) greater publication of incremental and, where appropriate, non-significant findings; (b) the use of increased power and greater sample sizes; (c) preregistration; (d) single-paper meta-analyses; (e) reporting all, rather than only significant, studies conducted; and, (f) data transparency—among many other recommendations [15–20]. Following such recommendations, studies retesting published findings and employing one or a combination of the proposed solutions are becoming commonplace [c.f., 21–23]. The present paper draws on such recommendations to reexamine the relation between optimism and counterfactual direction of comparison.

The goal of the present research is to conduct a new test of the relation between optimism and counterfactual direction of comparison using a methodological approach superior to what appears in the literature. We report the results of a large-sample, pre-registered main study, followed by a single-paper meta-analysis that includes six preliminary studies (see S1 File). Our meta-analytic summary of these data sets (7 total) provides a robust estimate of the effect size relating optimism to counterfactual direction of comparison.

## Theoretical background

Counterfactual thoughts play a key role in a range of emotions, judgments, and behaviors and have been applied in studies from moral judgment [24] to mental health [25] and from the neurocognitive underpinnings of choice [26] to the developmental progression of causal reasoning [27]. Counterfactuals are thoughts about the past, specifically how particular facets of the past may have been different from actuality. Such thoughts may contain kernels of insight that suggest new courses of action in the future. Indeed, the functional theory of counterfactual thinking asserts that much of counterfactual thinking is oriented toward the formation of intentions that embody learning and promote improvement [3, 28] and may therefore be useful for goal pursuit [3, 28–31].

According to the functional theory of counterfactual thinking, individuals tend to generate counterfactuals when there is a discrepancy between their actual state and desired end-state. And more specifically, counterfactual direction of comparison characterizes the sorts of counterfactuals that arise spontaneously in light of such a discrepancy. Upward counterfactuals, because they specify improvement to the status quo, may be useful in articulating means by which future performance might be improved. Indeed upward counterfactuals predominate in response to recognition of an actual-ideal discrepancy. Conversely, downward counterfactuals are related to affect regulatory goals. By way of a contrast effect, consideration of inferior outcomes can make factual outcomes seem more positive. As a result, positive emotions such as relief are evoked by downward counterfactual thoughts; such thoughts may thus be generated strategically to repair mood. Although downward counterfactuals are generated less frequently

overall than upward counterfactuals [32–35], they arise when individuals feel a need to compensate for negative emotional states [36].

The link between goals and counterfactual direction of comparison is further illuminated by consideration of the antecedents to counterfactual thinking. At the most basic level, outcome valence is a key antecedent, such that negative more than positive outcomes activate upward counterfactual thinking [37–39]. Additionally, situationally active performance goals differentially influence counterfactual direction, such that when goals remain active (e.g., for tasks that involve a repeating sequence) versus inactive (e.g., completed tasks), upward counterfactuals are generated more frequently [40]. In addition, individual differences in chronic goals may also be antecedents to counterfactual direction [41–43]. For example, incremental (vs. entity) theorists, who see human behavior as more variable and hence improvable, are more likely to generate upward (vs. downward) counterfactuals [43]. This same link is reflected in the "opportunity principle," whereby the opportunity to change or amend prior outcomes is associated with the generation of more upward (vs. downward) counterfactuals [3, 44]. In these varying ways, goal cognition constitutes a key determinant of counterfactual direction of comparison.

A fundamental question, then, centers on how individuals form cognitions of the past and future when it comes to counterfactual thinking, and whether there is meaningful variation across individuals in this intersection. One way that prior theory has addressed this question is via the relation between dispositional optimism and counterfactual direction of comparison [6–11]. Optimists expect that good things will occur, resulting in confidence and persistence in the face of challenges [45]. Moreover, optimism connects to models of behavioral self-regulation, in that people engage in goal-congruent efforts to the extent that they expect their efforts will eventually result in success [46, 47].

Dispositional optimism may therefore predict counterfactual direction. One way to define optimism is at a domain-general (vs. domain-specific) level, as instantiated by the Life Orientation Test (LOT and LOT-R; [5, 47]). Defined in this way, optimism is associated with superior coping outcomes [46, 48–51]. For example, coronary bypass patients who scored higher in optimism recovered more quickly and reported higher quality of life six months post-surgery [50]. Although optimism may engender positive coping outcomes partly via affect regulatory processes (e.g., a self-serving attributional basis that mitigates the affective sting of negative outcomes [52]), the bulk of recent evidence suggests that optimism brings beneficial outcomes via performance improvement goals. For example, Scheier et al. [47] showed that optimism more often involves performance improvement processes (e.g., active coping, planning, seeking social support) than affect regulation (e.g., denying or disengaging). Nes and Segerstrom's [53] meta-analysis ($N = 11,629$) indicated that optimism is associated more strongly with performance improvement goals (defined in terms of approach goals and active coping) than with affect regulation (defined in terms of avoidance goals and passive coping). To be sure, performance improvement is not the same as an approach goal, nor is affect regulation synonymous with avoidance, yet nevertheless the degree of conceptual overlap indicates an overarching connection between optimism and performance improvement. Upward counterfactuals connect to performance improvement goals, whereas downward counterfactuals connect to affect regulatory goals (e.g., [29, 32, 36, 40, 54, 55]). Thus, the optimism literature provides a basis for predicting, through shared conceptual emphasis on performance improvement, that optimism may predict upward (vs. downward) counterfactual thinking.

## Extant findings

In contrast to the theoretically derived prediction described above, the counterfactual literature indicates that optimism predicts downward (vs. upward) counterfactual thinking [6–11].

For example, Kasimatis and Wells [8] operationalized optimism with the LOT scale to predict counterfactual thoughts collected in a thought-listing task. Those higher in optimism generated more downward than upward counterfactuals, but Kasimatis and Wells' book chapter report omitted many procedural and statistical details. Sanna [10] operationalized optimism with the Defensive Pessimism Questionnaire (DPQ; [52]) to predict students' course-related counterfactual thoughts collected via thought-listing—again, those higher in optimism generated more downward than upward counterfactuals. However, in this as well as a follow-up study with similar findings [11], important statistical details were omitted, such as those centering on the main effect of optimism on counterfactual direction. Further, the use of tertile splits and exclusion of the middle third of participants from analyses raises questions of statistical precision in light of emerging data standards (e.g., [56]). Issues with dichotomizing the optimism measure [7] as well as omitted statistical analyses [6, 9] further impact the conclusions of the remaining papers to have shown that optimism predicts downward (vs. upward) counterfactual thinking. A final point of concern is that to date, eight of Sanna's papers have been retracted because of data fraud [57]. Although the two relevant papers cited here have not been retracted, a prudent reading of the literature suggests the need for a new look at the relation between optimism and counterfactual direction of comparison.

## Main study

To that end, our main study features four key improvements over the literature. First, we focus on episodic counterfactuals that participants report regarding their own autobiographical experiences [58]. The method of soliciting memories of episodic counterfactuals is superior to two other methods to elicit counterfactual thinking: a) hypothetical scenarios (used in [8]), a method vulnerable to the critique that participant responses are speculative than genuine, and b) laboratory task (used in [10]), a method that, although yielding more genuine responses, lacks cross-domain generality. Second, participants self-code the direction of each counterfactual they generate using scale ratings, preventing independent coders (as in [8, 9, 10]) from possible misinterpretation of ambiguous counterfactuals (e.g., "what if I had taken a 'different' train" could represent either a better or a worse alternative). Moreover, although much prior research has dichotomized direction into upward versus downward, some prior research has used scales (e.g., [59]), with the advantage of capturing greater variability in the way individuals report on counterfactual thoughts. Third, we assessed optimism using the LOT-R [47], perhaps the best validated and most widely used of available optimism measures. Fourth, the main study method solicited four episodic counterfactuals from each participant in an attempt to enhance reliability of measurement via an entirely within-subject design.

The main study assessed the magnitude and direction of the association between optimism and counterfactual direction of comparison. We report all conditions, measures, and any data exclusions. The IRB of Northwestern University, Kellogg School of Management approved this study for Human Subject Research. Written informed consent was obtained from all participants prior to their commencing the study and before any data collection. The study was pre-registered on 10-9-2018, and all materials and de-identified data are available at: https://osf.io/wudjs/.

We hypothesized that optimism would not predict greater downward (vs. upward) counterfactual thinking. Further, based on evidence from the counterfactual and optimism literatures connecting both upward counterfactuals and more optimistic individuals to improved performance on goals, we theorized the opposite pattern may emerge—that is, that optimism may predict greater upward (vs. downward) counterfactual thinking. Thus, our aim was to

reexamine the link between optimism and counterfactual direction of comparison that has received prior research attention.

## Method

**Sample size and power.** An a priori power analysis was conducted using G*Power v3.1.9.4 to determine the minimum sample size required to find significance based on an effect size of = .08, alpha = .05, power = .8, two-tail t-test, using a within-subject design. This analysis resulted in a desired sample of 1,229 participants. Although this power calculation was based on a t-test, our analysis relies primarily on a mixed effects model (MEM). Given continued debate on how best to calculate power for MEMs, and given such models should be more sensitive than t-tests, we believe that this power calculation is a reasonable way to determine sample size in our study. Specifically, sample size requirements for the analyses for the predictors in our model (optimism, context, and opportunity) would fall within this minimum sample size of 1,229 [60–62]. From our prior experience with attrition rates, we assumed a completion rate of .6 from Time 1 (T1) to Time 2 (T2) and on this basis we set the desired sample size for T1 at $N$ = 2,050.

The final sample (comprising participants who completed both T1 and T2) consisted of 1,150 adults drawn from MTurk (59% female; $M_{age}$ = 36, $SD_{age}$ = 12). The T1 sample consisted of 2,059 adults; out of this number 1,287 (63%) returned at T2. Participants not following instructions (e.g., answered "na" or gibberish, n = 119) or failing attention checks (n = 18), as determined by the first author, were excluded from analyses (total exclusions, n = 137). T1 data collection took place between 10-4-2018 and 10-5-2018; T2 data collection took place between 10-11-2018 and 10-14-2018.

**Research design and measures.** The T1 assessment focused on optimism, measured using the 6-item LOT-R with 5-point response options, averaged to create the optimism index ($\alpha$ = .88). Participants responded to demographic questions (e.g., age, gender, race, and ethnicity) and an open-ended attention check ("Please confirm you are human by describing the weather outside right now where you are").

The T2 assessment centered on counterfactual thinking. To introduce variability in the tendency to report upward and downward counterfactuals, participants reported counterfactual alternatives to four autobiographical events with prompts that varied according to a 2 (context: personal vs. professional) × 2 (opportunity: low vs. high) factorial design (fully within-subject). Participants read the instructions: "This survey asks you about four separate events from your recent past. Your job is to answer brief questions about what you remember." Participants then responded to four prompts: 1) low opportunity professional, 2) low opportunity personal, 3) high opportunity professional, and 4) high opportunity personal (with order of presentation randomized). For each, participants recalled and shared details about a recent negative experience (see Table 1). The choice of personal versus professional context was based on the

**Table 1. Main study prompts by condition.**

|  | Professional | Personal |
|---|---|---|
| **High Opportunity** | Think back to a recent NEGATIVE experience you had at WORK or SCHOOL that there are POSSIBLE SOLUTIONS TO. | Think back to a recent NEGATIVE experience you had with FRIENDS or FAMILY that there are POSSIBLE SOLUTIONS TO. |
| **Low Opportunity** | Think back to a recent NEGATIVE experience you had at WORK or SCHOOL that there is NO WAY TO RESOLVE. | Think back to a recent NEGATIVE experience you had with FRIENDS or FAMILY that there is NO WAY TO RESOLVE. |

Exact prompts used in Main Study assigning participants to the 2 (context: personal vs. professional) × 2 (opportunity: low vs. high) factorial design (within-subject) conditions.

observation that life regrets, which derive from counterfactual thinking, tend to focus on these contexts more frequently than others [44, 63]. The opportunity manipulation derived from past demonstrations that upward counterfactuals are more common under conditions of high (vs. low) opportunity [3]. The manipulation of opportunity thus afforded an internal, theoretically-based check on our measurement technique. Replication of this known effect lends credibility to our methods.

Next, we solicited counterfactual thoughts with a prompt that intended to be neutral with regard to upward or downward direction of comparison: "After having experiences like this, sometimes people have thoughts like 'what if'—in that they think about how things could have gone differently. In the space below, please share one 'what if' thought."

Our dependent measure, counterfactual direction of comparison, was assessed using a self-report rating scale. Specifically, after each counterfactual prompt, participants responded to: (1) "Does your 'what if' thought focus more on how things could have gone better or how things could have gone worse?"; (2) "In general when you look back at this experience, do you tend to ponder more about how things could have gone better or how things could have gone worse?"; and, (3) "When you look back at this experience, does it make more sense to you to reflect on how the outcome could have been better or how the outcome could have been worse?" on 5-point scales anchored at [-2] = *Definitely Worse* to [2] = *Definitely Better*. We averaged these three items to create an index of counterfactual direction for each of the four prompts ($\alpha_1 = .87$; $\alpha_2 = .88$; $\alpha_3 = .90$; $\alpha_4 = .90$; overall $\alpha = .85$). Participants responded to the same demographic measures and attention check as in T1.

## Results

Analyses were conducted using JMP Pro v14.1.0. In the counterfactual direction index, positive values indicate an upward direction of comparison and negative values indicate a downward direction of comparison. Overall, the mean counterfactual direction score was greater than zero, $M = 0.98$, $SD = 0.75$; $t(1149) = 44.33$, $p < .001$, 95% CI [0.96, 1.00], revealing a general tendency toward generating more upward than downward counterfactuals, an effect consistent with the prior literature (e.g., [32–35]).

We ran a mixed regression model with mean-centered optimism ($M = 2.28$, $SD = 0.88$), context (professional = 0; personal = 1), opportunity (low = 0; high = 1), and all 2 and 3-way interactions between these factors, as predictors of counterfactual direction, with participant-level variation accounted for as a random effect. This overall model was significant, AICc = 13495.93, $p < .001$. Critically, this regression showed a significant effect of optimism, $b = .06$, $SE = .03$, $\beta = .05$, $t(1152.5) = 2.29$, $p = .022$, 95% CI [.008, .11], such that greater optimism predicted more upward counterfactuals (see Fig 1). The main effect of context was not significant, $b = -.01$, $SE = .01$, $\beta = -.02$, $t(3447.9) = -1.08$, $p = .26$, 95% CI [-0.04, 0.01]. However, we did observe a significant main effect of opportunity, $b = .07$, $SE = .01$, $\beta = .07$, $t(3447.9) = 5.32$, $p < .001$, 95% CI [0.05, 0.10], such that events high (vs. low) in opportunity predicted more upward counterfactuals. Because this opportunity effect replicates prior research [3], we gain confidence in the fidelity of the method. There were no significant 2-way or 3-way interactions ($ps > .23$).

In support of our hypothesis, the main study reveals that optimism does not predict greater downward counterfactual thinking, as prior research had found. Instead, this study shows that optimism weakly predicts greater upward than downward counterfactual thinking and that this effect is consistent across two different contexts of life, personal and professional.

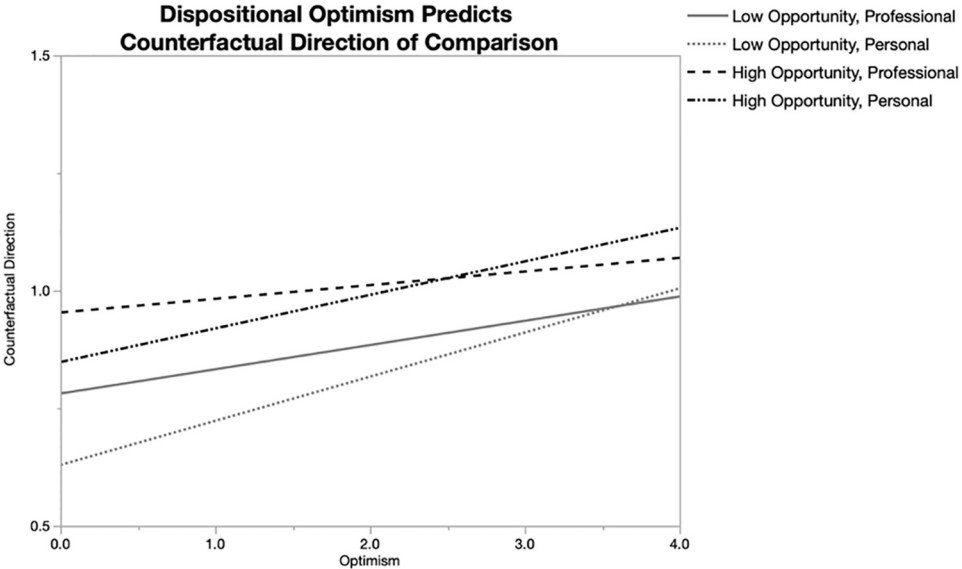

**Fig 1. Dispositional optimism predicts counterfactual direction of comparison, moderated by context and opportunity.**

## Statistical summary of studies

We ran six preliminary studies between February 2017 and March 2018. We summary these prior studies here to avoid a potential bias by not reporting prior unpublished studies and to illuminate the rationale for the main study's sample size. We report all methodological details of the preliminary studies in the S1 File. Further, Table 2 summarizes key methodological details (measures and manipulations) as well as the results, focusing on the effect (β) of optimism on counterfactual direction of comparison.

Table 3 summarizes the focal relation between optimism and counterfactual direction of comparison as estimated across all studies conducted. We conducted a meta-analysis as per McShane and Bockenholt [19] by way of 1) averaging across all conditions within a study, 2) computing the correlation between the two key variables across each study, 3) converting that correlation to the Fisher $Z$ scale, and 4) analyzing via the basic random effects meta-analytic model (see S1 File for R-code). From this analysis ($N$ = 3,051), we noted a mean effect size of $r = 0.06$, $SE = .02$, $Z = 3.38$, $p = .0007$, 95% CI [.03, .10]. Importantly, although this effect is weak, it supports our initial hypothesis that optimism does not predict greater downward counterfactual thinking. Instead, this meta-analysis suggests optimism is more clearly, albeit weakly, linked to greater upward counterfactual direction of comparison.

**Table 2. Summary of methodologies and results across all studies.**

| Study | N | Optimism Measure(s) (IV) | Counterfactual Direction Measure (DV) | Moderators (Manipulated) | Effect (β) |
|---|---|---|---|---|---|
| P1 | 197 | LOT-R | Three-item scale | Outcome Valence | β = 0.07, *p* = .46 |
| P2 | 494 | Same as P1 | Dichotomous | Same as P1 | β = 0.15, *p* = .29 |
| P3 | 199 | Same as P1 | Same as P1 | | β = 0.02, *p* = .80 |
| P4 | 290 | Same as P1 | Same as P1 | Same as P1 | β = 0.10, *p* = .09 |
| P5 | 196 | Same as P1 and DPQ | Same as P1 | Same as P1 | LOT-R: β = 0.07, *p* = .43 DPQ: β = -0.07, *p* = .47 |
| P6 | 525 | Same as P1 and DPQ | Same as P1 | Same as P1 | LOT-R: β = 0.06, *p* = .15 DPQ: β = 0.06, *p* = .17 |
| Main | 1150 | Same as P1 | Same as P1 | Context, Opportunity | β = 0.05, *p* = .02 |

**Table 3. Statistical summary of studies.**

| Study | N | r | P |
|---|---|---|---|
| Preliminary 1 | 197 | 0.10 | 0.17 |
| Preliminary 2 | 494 | 0.04 | 0.35 |
| Preliminary 3 | 199 | -0.03 | 0.64 |
| Preliminary 4 | 290 | 0.06 | 0.28 |
| Preliminary 5 | 196 | 0.13 | 0.08 |
| Preliminary 6 | 525 | 0.06 | 0.15 |
| Main Study | 1150 | 0.07 | 0.02 |
| **Meta-analysis** | **3,051** | **0.06** | **.0007** |

Overall mean effect of optimism on counterfactual direction, as estimated across the six preliminary and one main studies.

## Conclusions

Is an optimist more likely to see counterfactual alternatives that specify a better (upward) or worse (downward) state of affairs, relative to actuality? This question hinges on their underlying goals, which might either center on performance improvement or affect regulation. If performance improvement goals dominate, the optimist will generate counterfactuals that help them to improve in the future (upward counterfactuals). But if affect regulatory goals dominate, the optimist will generate counterfactuals that help them to feel better in the moment (downward counterfactuals). The prior counterfactual literature indicates the latter answer, that optimism predicts downward counterfactual thinking. However, the theoretical consensus in the optimism literature suggests a different pattern, that optimism predicts upward counterfactual thinking. Given uncertainty surrounding the counterfactual literature (methodological, statistical, data reporting), we conducted new research to examine this relation, and provide evidence that optimism weakly predicts upward counterfactual thinking. Thus, our current result is consistent with the optimism literature (generally speaking) but not the counterfactual literature (as it pertains to optimism). Future research might explore potential moderators leading to our result compared to prior findings, potentially exploring the methodological differences as a factor. Furthermore, although we found a weak relation between optimism and upward counterfactual direction of comparison, future research may explore the role of performance improvement goals as linking these constructs.

Our key result does connect to a broader theme in the counterfactual literature, namely that "counterfactual thoughts often reflect goals and the varying means to reach those goals . . . Imagining alternative pathways by which past goals might have been achieved provides insights that comprise blueprints for future action" ([3], pp. 5). Our results thus speak to the intersection of past-focused versus future-focused thinking. As individuals look to the past to imagine alternatives to factual events, they likely rely upon the same brain system (e.g., [26]) as when they look to the future to imagine those possibilities that may yet come to pass. Optimism, as an enduring individual difference, plays a role in thoughts of both the past and future.

## Supporting information

**S1 Appendix. Summary table of prior research.** A summary table of nine studies across six papers published between 1995 and 2015, which indicate that optimism predicts downward (vs. upward) counterfactual thinking. This summary table reports the authors and year of

publication; the study number within the publication (if applicable); the total sample size of the study (if reported); the design of that study including the conditions participants were assigned to or whether the design was correlational; how counterfactuals were elicited (i.e., in response to what prompts or events); the scale that was used to capture trait optimism; what optimism was compared to (if applicable); and, how counterfactuals were classified as downward or upward.
(DOCX)

**S1 File.**
(DOCX)

## Acknowledgments

We thank Richard Robins and Suzanne Segerstrom for their comments on an early manuscript draft; and Daniel Jung, Yiyun Lan, Jue Wu, and Michelle Zhou for assistance in data collection and library search.

## Author Contributions

**Conceptualization:** Jessica Gamlin, Rachel Smallman, Kai Epstude, Neal J. Roese.

**Data curation:** Jessica Gamlin.

**Formal analysis:** Jessica Gamlin, Neal J. Roese.

**Funding acquisition:** Neal J. Roese.

**Investigation:** Jessica Gamlin, Neal J. Roese.

**Methodology:** Jessica Gamlin, Rachel Smallman, Kai Epstude, Neal J. Roese.

**Project administration:** Jessica Gamlin.

**Supervision:** Jessica Gamlin, Rachel Smallman, Kai Epstude, Neal J. Roese.

**Visualization:** Jessica Gamlin.

**Writing – original draft:** Jessica Gamlin, Neal J. Roese.

**Writing – review & editing:** Jessica Gamlin, Rachel Smallman, Kai Epstude, Neal J. Roese.

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
