## [Decision Letter · Decision Letter 0]

3 Apr 2020

PONE-D-20-00918

Does dispositional optimism predict counterfactual direction of comparison?

PLOS ONE

Dear Dr. Gamlin,

Thank you for submitting your manuscript to PLOS ONE. After careful consideration, we feel that it has merit but does not fully meet PLOS ONE’s publication criteria as it currently stands. Therefore, we invite you to submit a revised version of the manuscript that addresses the points raised during the review process.

We would appreciate receiving your revised manuscript by May 18 2020 11:59PM. To enhance the reproducibility of your results, we recommend that if applicable you deposit your laboratory protocols in protocols.io, where a protocol can be assigned its own identifier (DOI) such that it can be cited independently in the future. For instructions see: http://journals.plos.org/plosone/s/submission-guidelines#loc-laboratory-protocols

We look forward to receiving your revised manuscript.

Kind regards,

Peter Karl Jonason

Academic Editor

PLOS ONE

2. Please consider changing the title so as to meet our title format requirement (https://journals.plos.org/plosone/s/submission-guidelines). In particular, the title should be "Specific, descriptive, concise, and comprehensible to readers outside the field" and in this case it is not informative and specific about your study's scope and methodology.

3. Please provide additional details regarding participant consent. In the ethics statement in the Methods and online submission information, please ensure that you have specified whether consent was informed.

Reviewers' comments:

Reviewer's Responses to Questions

**Comments to the Author**

1. Is the manuscript technically sound, and do the data support the conclusions?

Reviewer #1: Partly

Reviewer #2: Partly

2. Has the statistical analysis been performed appropriately and rigorously? 

Reviewer #1: No

Reviewer #2: Yes

3. Have the authors made all data underlying the findings in their manuscript fully available?

Reviewer #1: Yes

Reviewer #2: Yes

4. Is the manuscript presented in an intelligible fashion and written in standard English?

Reviewer #1: Yes

Reviewer #2: Yes

5. Review Comments to the Author

Reviewer #1: The authors present a large-scale study, as pilot studies, aiming to assess the relation between trait optimism and counterfactual thinking focusing on upward vs. downward comparisons. This is clearly a considerable research effort, testing many participants over time, which should hence be published to be available to interested researchers. However, I have several major concerns regarding the methods and statistics, which question the appropriateness and relevance of the conclusions. The questions with the methods used, and the apparent very small size of the effect of interest seems to significantly question the relevance, generalizability, and replicability of such effects. The authors should recognise and discuss this clearly in the article, and further considering and discussing how it fits more broadly in the literature would increase the contribution of this work to the field.

- The statistical analyses across the paper are confusing, often insufficiently described, and/or can seem inappropriate. The variability in the statistical analyses (i.e. regressions, correlations) used within and across studies to test the same hypothesis is confusing, and hinders an overall understanding of the strength of evidence and estimating effect sizes for the hypotheses of interest.

- Power calculation – the rationale for this is not clear, and does not seem appropriate. It was calculated for a t-test, based on an unspecified and unjustified effect size estimate, without clarifying what hypothesis that would concern. More importantly, this is not related to the actual statistical tests used to assess the relevant hypotheses, i.e. regression models, nor the reported effect sizes. Hence, it is not pertinent to assessing the power of the analyses actually used (and standard power calculators do offer the possibility to calculate power for a regression model or a correlation). This is a common issue, and the past cannot be changed, and clearly the most important is that some method was used to estimate a target sample size in advance of analysis. Nonetheless, the authors should be clear about what they actually did and what was their rationale, as well as account for the divergence in the analyses.

- The description of the “single-factor” regression and “mixed regression” models (p.11-12) is insufficient. What software and methods more exactly were used to estimate the models and the reported parameters (e.g. CIs, degrees of freedom, p-values)? What is the rationale and meaning of inconsistently reporting different effect sizes across, e.g. eta squared, d… Unclear whether the predictor optimism was mean centred, which is recommended so the remaining parameters are estimated at that average level. It’s unclear why the authors run a single-factor, and then mixed regression. If the other predictors are plausible modulators, that single-factor model seems pretty meaningless, and it is also not serving the purpose of replicating a similar analysis from other studies, since later correlations are used. It’s also unclear to me how the “mixed regression model” was specified. The authors state their DV is the ratings across 3 questions, so if that average was used in the model, and all other effects are between subjects, then it’s unclear what data would be nested within “subject” (hence why use mixed regression). If they include the 3 measurements separately for each participant (which would be best), then in fact the role of the question itself should be modelled as crossed design, a separate “random” effect (e.g. often called “item” effects, cf. Baayen et al 2008; Bates et al 2015; Barr et al 2013). The full results table of the tested models should also be included in the article. [Note that, to avoid the inconsistency in tests within and across studies, could use mixed effects models for a meta-analysis of all the data, using the study as a higher-level, nesting factor. There obviously can be good reasons for choosing other methods, but might be worth thinking about whether the relevant tests and statistics should focus on correlations or on regression.]

- In fact, I’m not clear on the rationale for averaging across the 3 questions about counterfactuals that concern different issues. Whether trait optimism is related to beliefs about counterfactual thinking vs. actual behaviour while engaging in it are actually two different questions. But currently the subjective ratings to the 3 questions posed currently confuse those two aspects. The authors could in fact analyse that separately. The authors also did not comment on whether the counterfactuals produced by the participants, if rated by an external observed, would indeed be in line with the subjective reports. In other words, the trait vs. state production of counterfactuals seems confused by the design and the measurements, and the article doesn’t clearly address that point.

- Looking at the reported meta-analysis does not yield much confidence in the overall conclusions, since only the final large study would seem to robustly show the hypothesised effect. While the much larger sample in the final study obviously yields more reliable evidence, hence likely yielding an overall effect, this also seems to question what the relevance of such a small effect is, if it can’t be easily reproduced, even when the previous studies had relatively large samples (N>200).

- The authors highlight throughout how one’s goals are key mediators of whether up/downward counterfactuals will be produced, but their main analyses do not actually address that moderator. The predictions about the role of goals also seem to predict interactions in quite opposing directions, e.g. more upwards counterfactual if aiming to improve performance + have opportunity to change future outcomes, but downward counterfactuals to regulate mood when there’s nothing you can do about it. While they find that opportunity is related to higher ratings (more likely upward) than “no opportunity”, both are quite clearly in upward side of the scale. As the interaction with outcome valence is described, one might predict that is likely because they chose to only use scenarios with negative outcomes, but then that seems to limit the scope of the conclusions about the relation between optimisms and counterfactuals, if the scenarios are already likely to yield upwards comparisons?

- The supplemental materials describing the previous studies should be improved to more clearly summarise the information of what was varied across the studies and what the various results would be (maybe akin to the Appendix table, but more relevant to summarising the relevant points). Including figures/tables summarising the effects in the regression models would also help the reader.

- Appendix – Should clarify the methods involved in collecting this list, the meaning of the columns, etc, and its purpose here… How were the sample size estimates obtained? Is there actually no info on the N per group in any of the studies? Were all manipulations are always done between participants?

- The authors vaguely state that there has been controversy in the literature, and that there are flaws and limitations in the previous work, which is understandably a good argument for make a robust new study. But they could more clearly address what that should actually imply for how to interpret the previous literature, and where/whether there might be reasonable methodological differences, e.g. related to the moderators mentioned, or whether the measures target beliefs vs. behaviour, which could explain opposing patterns of relations with optimism. Such a more detailed discussion could well be moved to the appendix, which could currently be mostly puzzling for someone who might not already be familiar with the details of that work.

- The 4th paragraph seems to basically repeat what was said in the 2nd, while possibly expanding on some points, but this should be combined to avoid repetition.

Reviewer #2: The manuscript explores and interesting question about the relationship between dispostional optimism and the direction of comparison of counterfactual thoughts. I find the topic very important, bcause the forms of counterfactual thoughts might have consequences for emotion regulation. The introduction is generally well-written and the justification of the study is sufficient. However, I do not understand why the authors decided to report fully only one study out of seven? I believe the article can make much stronger contribution if the authors would decide to develop it into a full paper. I present my concerns below.

p. 9. Method - I am not sure for what kind of analysis the sample size has been determined? Is this sample to detect an effect of .08 in t-test comparison?

Unfortunately, the content of the part "Statistical summary of studies" is unclear. First, this should presented before their main study, because these are "preliminary" studies as the authors wrote. Second, I am not sure what is actually presented here. What do the authors mean that they try to "avoid file drawer issues"? As I understand, the authors conducted six other studies before the one reporting here. In this case I would encourage them to report them in the current manuscript. In the current version it is really confusing what are all those effects etc. This would strenghten their main finding. At least, I would suggest to present the summary of previous studies more clearly and earlier in the manuscript.

6. PLOS authors have the option to publish the peer review history of their article (what does this mean?). If published, this will include your full peer review and any attached files.

Reviewer #1: No

Reviewer #2: No

---

## [Author Response · Author response to Decision Letter 0]

23 Jun 2020

Response to Reviewer #1 (R1)

1. The bulk of R1’s comments centered on the statistical analysis, which we have substantially revised to provide additional clarity. Specifically:

a. We report the software and methods used and clarify how the mixed regression model was specified (page 12).

b. We mean-center the predictor, optimism, in our Main Study, as we had in our Preliminary Studies. 

c. Rather than reporting effect sizes, we report Standardized Beta and 95% Confidence Intervals for analyses pertaining to our central relation of interest (i.e., the effect of optimism on counterfactual direction of comparison).

d. We eliminate our single-factor regression in the main study.

e. We add further rationale supporting our decisions regarding the elicitation of our dependent measure (pages 7-8). We also explain why we decided to average across the three questions about counterfactuals (rather than running a random effects model assessing item effects to capture these separately). In short, prior research has used scales, with the advantage of capturing greater variability in the way individuals report on counterfactual thoughts. Moreover, our central aim with this measure was to capture a broad tendency (whether behavioral or belief-based) to think about a past outcome as it could have gone better or worse. Thus, averaging the three-items, we feel, best captures our intended construct of counterfactual direction of comparison. 

f. We enhanced our description of the original power analysis to better conform to reporting standards (page 9). 

2. R1 also requested independent coders for the counterfactuals (rather than using participant’s self-reported coding of upward or downward). As we now elucidate in the manuscript (page 8), independent coding suffers when ambiguous counterfactuals arise. Self-coding prevents misinterpretation by independent coders—e.g.,:

a. One participant discussed their negative professional experience, “We're now forced to use an electronic medical record, but it is not made to be used in my particular field,” and generated a counterfactual as follows, “What if they had considered my field when shopping and found a different system that met our needs?” It is impossible to know whether the “different system” referred to constitutes a better or worse system, and thus it is impossible for an independent coder to reliably code the direction of this counterfactual. 

b. Similarly, another participant noted, “My boss is an idiot, doesn't understand English, and is going deaf. What if I worked for a different company?” Again, an independent coder cannot reliably code the direction for such a counterfactual. 

Such examples are sufficiently common in this dataset, making independent coding an unreliable source for assessing the directionality of counterfactuals and suggesting that self-coded counterfactuals (as is also standard practice in the counterfactual literature) is a robust method for capturing counterfactual direction of comparison. 

3. R1 suggested that our meta-analytic conclusion “does not yield much confidence in the overall conclusions.” We agree that, on the one hand, a meta-analysis showing a weak correlation between optimism and upward counterfactuals, and only significant when sample sizes are large (i.e., N > 1,000), does not provide evidence of a strong link between optimism and upward counterfactual direction of comparison. On the other hand, this is precisely why our meta-analytic approach is appropriate. Our goal in this paper, as is now stated earlier in the text for enhanced clarity (pages 3, and 8-9), is to correct a long-held belief in the literature that optimism is robustly linked to downward counterfactual direction of comparison. The purpose of our meta-analysis is to support a revised conclusion that there is not a robust link between optimism and upward counterfactual direction of comparison and only “a weak relation” between optimism and upward counterfactual thinking (page 14). This is our critical finding and the primary contribution of our paper. 

4. Relatedly, R1 suggests we explore mediation by goals in more depth. While we agree that this is a worthwhile pursuit, we believe it is outside the scope of this initial investigation. Specifically, we emphasize that the main finding and primary contribution of our paper is to correct a conclusion that has persisted in the literature for decades. We more carefully emphasize this in the introduction (page 3) and suggest future directions pertaining to this suggestion by R1 (page 15). 

5. Also, relatedly, R1 suggests adding more clarity around the limitations of the previous work, what this implies for how to interpret the previous literature, and where/whether there might be reasonable methodological differences that account for the differences. We have edited the text to bring enhance clarity on this matter (pages 7-8) and additional suggest future work that might investigate this in more depth (page 15). 

6. R1 notes that restricting episodic recall in our main study to include only negative events could limit our conclusions on the relation between optimism and counterfactual direction of comparison. To clarify, the prior literatures shows that counterfactual thoughts are much more likely to be spontaneously considered after a negative than after a positive event, and indeed the frequency of counterfactual thoughts after positive events is so low that meaningful variation is absent. Moreover, the literature also shows a robust main effect of more upward (vs. downward) counterfactuals in response to negative outcomes. However, there is nothing to suggest that optimists recall positive and negative events in a way that would differentially affect the generation of counterfactuals. Thus, omitting the recall of positive events has minimal impact on the broader correlation that we find between optimism and upward counterfactual direction of comparison. Thus, we feel limiting our main study to not include an additional factor of valence is warranted. 

7. Finally, R1 had some minor comments, which we address as follows:

a. More clearly summarizing the information of what varied across the preliminary studies and the results (see new Table 2). 

b. Streamlining the table in the appendix and adding clarification for each of the columns’ meaning. 

c. Revising the writing throughout so as to avoid redundancies.

Response to Reviewer #2 (R2)

R2 had two main points:

1. Questioning our rationale for reporting only 1 out of 7 studies, and, relatedly, suggesting we present the statistical summary of studies prior to our main study

a. With respect to including all studies in the main text, we felt that this would not be appropriate given we intended them to be preliminary studies. Our rationale for including them in the Supplemental Materials was primarily one of full data transparency, but also to provide a clear summary of the different methods we used that might potentially replicate the effect in the prior literature that optimism was associated with more downward counterfactuals. We also included them to show that this effect was not replicated, providing important evidence to justify our need to run a sufficiently-powered main study examining the relationship between optimism and counterfactual direction of comparison. However, we do now include a new table (Table 2) summarizing exactly what varied methodologically across the preliminary studies and the results (focusing on β to facilitate comparison across studies). 

2. Asking for more clarity on how we determined sample size and the accompanying power analysis. 

b. We enhanced our description of the original power analysis to better conform to reporting standards (page 9).

---

## [Decision Letter · Decision Letter 1]

16 Jul 2020

PONE-D-20-00918R1

Dispositional optimism weakly predicts upward, rather than downward, counterfactual thinking: A prospective correlational study using episodic recall.

PLOS ONE

Dear Dr. Gamlin,

Thank you for submitting your manuscript to PLOS ONE. After careful consideration, we feel that it has merit but does not fully meet PLOS ONE’s publication criteria as it currently stands. Therefore, we invite you to submit a revised version of the manuscript that addresses the points raised during the review process. As you will read below, you will see that both reviewers were in favor of your paper being published, congratulations. Despite this, there is room for improvement that I hope you can address easily and quickly to get your paper published in PLOS 1. Some areas of concern are (1) clarifying the power calculations, (2) ensuring the OSF site is populated and prepared properly, (3) and better highlighting the importance of this study. After addressing these issues, I suspect the paper will be ready to be accepted.

We look forward to receiving your revised manuscript.

Kind regards,

Peter Karl Jonason

Academic Editor

PLOS ONE

Reviewers' comments:

Reviewer's Responses to Questions

**Comments to the Author**

1. If the authors have adequately addressed your comments raised in a previous round of review and you feel that this manuscript is now acceptable for publication, you may indicate that here to bypass the “Comments to the Author” section, enter your conflict of interest statement in the “Confidential to Editor” section, and submit your "Accept" recommendation.

Reviewer #1: (No Response)

Reviewer #2: All comments have been addressed

2. Is the manuscript technically sound, and do the data support the conclusions?

Reviewer #1: Yes

Reviewer #2: Yes

3. Has the statistical analysis been performed appropriately and rigorously? 

Reviewer #1: Yes

Reviewer #2: Yes

4. Have the authors made all data underlying the findings in their manuscript fully available?

Reviewer #1: No

Reviewer #2: Yes

5. Is the manuscript presented in an intelligible fashion and written in standard English?

Reviewer #1: Yes

Reviewer #2: Yes

6. Review Comments to the Author

Reviewer #1: The authors have made a clear effort to address the reviewers’ comments, which I believe has greatly improved the manuscript. But I believe the following minor points should be revised before publication.

The information provided for the power calculation is very vague to allow others to replicate (e.g. “weak effect”?). There’s still much debate concerning how to calculate power for mixed effects models, plus mentioning “paths” when running MEMs (and not SEMs) seems quite confusing. As the authors linked to their OSF page, despite the unclear labelling of document names, I found that the justification provided there in “Sample Size 2.rtf" is more precise, and clarifies that was run for a t-test. Since that’s probably the actual basis for their sample size, I suggest they copy that text to the manuscript, but acknowledge/explain that is not the power of the statistics actually used. I’d think that’s a plausible way for deciding on sample size, given challenges with assessing power for MEMs, and MEMs should actually be more sensitive and robust than t-tests. I just think it’s important to be accurate and transparent about what was/is done.

The OSF page linked currently doesn’t show any files in the “dataset” folder. There also seems to be some things that imply there was a 7th study, and the current main study would be number 8. Adding some clarification notes on OSF on the data/studies included there and pointing people to the relevant files would help to avoid confusion.

Regarding the MEMs analysis/results, the authors should report the method used for calculating degrees of freedom, and those should be reported with the remaining test statistics (i.e. the associated p values).

Reviewer #2: All comment have been addressed.

7. PLOS authors have the option to publish the peer review history of their article (what does this mean?). If published, this will include your full peer review and any attached files.

Reviewer #1: No

Reviewer #2: No

---

## [Author Response · Author response to Decision Letter 1]

29 Jul 2020

Response to The Editor

The Editor has asked us to better highlight the importance of this study. We now include a clear summary justifying the importance of this study in the “Goal of the present research” section of the paper. [See: pages 3-4] 

Response to Reviewer #1 (R1)

We are thankful for R1’s careful attention to detail and for guidance that has improved our paper. We have updated our manuscript to address the following concerns raised by R1:

1. Clarify the Power calculation. We updated the manuscript to be not only consistent with the OSF preregistration, but also more transparent vis-à-vis the general debate on how to best calculate power for MEMs (which R1 rightly highlighted). [see: Pages 9-10] 

2. Ensure the OSF site is prepared properly. Our OSF page (https://osf.io/wudjs/?view_only=389862820cdb43bdaaa6bda477202a48) now includes (a) all data files, (b) clearer file naming and folder organization, and (c) a folder entitled “0. Information (read first)” containing a text file explaining the components included in the OSF page. Specifically, that file reads:

“This OSF project contains the following documents:

1. STUDY DESIGN. 

- Basic information about how Preliminary Study 6 and the Main Study were designed and planned is included in text files. 

- Study Design information was uploaded prior to data collection being completed or any data analysis took place.*

2. SAMPLE SIZE. 

-Basic information on how the sample size was justified (Preliminary Study 6 and Main Study) and calculated is included. 

- Sample Size justification information was uploaded prior to data collection being completed or any data analysis took place.*

3. SURVEYS. 

- Surveys have been added in both .qsf (Qualtrics) and .docx (Microsoft Word) format. The .qsf files should be able to be uploaded directly to Qualtrics for replication. Qualtrics surveys can be shared directly (i.e., from Qualtrics account to Qualtrics account) by the first author upon request. Surveys from all studies are available upon request from the first author in either .qsf or .docx format. 

- Surveys were uploaded after data collection and analysis was complete. 

4. DATA. 

- All data files: Preliminary Study 1-6 and the Main Study. 

- Data files were uploaded after data collection and analysis was complete. 

*Due to an error in the timing of the launch of the Time 1 Main Study survey, the sample size justification and study design files were uploaded after the survey had been launched, but before looking at, downloading, or analyzing any data. Note that these files were however uploaded prior to the launch of the Main Study - Time 2 Survey, and therefore uploaded prior to any data merging, cleaning, or analysis taking place.”

3. Include t-test and degrees of freedom in Main Study. We had replaced the t-value and corresponding DF from our 2nd round submission with the standard beta and corresponding confidence interval. We are happy to also report the t-test and corresponding DF, and have added this information back into the results of the Main Study. [See: page 13]

Response to Reviewer #2 (R2)

We are thankful for R2’s support of our manuscript. As R2 did not raise any specific points for further clarification in this round of revision, we have not made any specific changes for R2.

---

## [Editor Report · Decision Letter 2]

31 Jul 2020

Dispositional optimism weakly predicts upward, rather than downward, counterfactual thinking: A prospective correlational study using episodic recall.

PONE-D-20-00918R2

Dear Dr. Gamlin,

We’re pleased to inform you that your manuscript has been judged scientifically suitable for publication and will be formally accepted for publication once it meets all outstanding technical requirements.

Kind regards,

Peter Karl Jonason

Academic Editor

PLOS ONE
---

## [Editor Report · Acceptance letter]

6 Aug 2020

PONE-D-20-00918R2 

Dispositional optimism weakly predicts upward, rather than downward, counterfactual thinking: A prospective correlational study using episodic recall. 

Dear Dr. Gamlin:

I'm pleased to inform you that your manuscript has been deemed suitable for publication in PLOS ONE. Congratulations! Your manuscript is now with our production department. 

Kind regards, 

on behalf of

Dr. Peter Karl Jonason 

Academic Editor

PLOS ONE